# Waste Fiber-Based Poly(hydroxamic acid) Ligand for Toxic Metals Removal from Industrial Wastewater [note 1]

**DOI:** 10.3390/polym13091486

**Published:** 2021-05-06

**Authors:** Md. Lutfor Rahman, Zhi-Jian Wong, Mohd Sani Sarjadi, Collin G. Joseph, Sazmal E. Arshad, Baba Musta, Mohd Harun Abdullah

**Affiliations:** 1Faculty of Science and Natural Resources, Universiti Sabah Malaysia, Kota Kinabalu 88400, Malaysia; javentwong@gmail.com (Z.-J.W.); msani@ums.edu.my (M.S.S.); collin@ums.edu.my (C.G.J.); sazmal@ums.edu.my (S.E.A.); babamus@ums.edu.my (B.M.); harunabd@ums.edu.my (M.H.A.); 2Seaweed Research Unit, Faculty of Science and Natural Resources, Universiti Malaysia Sabah, Kota Kinabalu 88400, Malaysia

**Keywords:** adsorption, waste fiber, poly(hydroxamic acid), wastewater, heavy metals

## Abstract

Toxic metals in the industrial wastewaters have been liable for drastic pollution hence a powerful and economical treatment technology is needed for water purification. For this reason, some pure cellulosic materials were derived from waste fiber to obtain an economical adsorbent for wastewater treatment. Conversion of cellulose into grafting materials such as poly(methyl acrylate)-grafted cellulose was performed by free radical grafting process. Consequently, poly(hydroxamic acid) ligand was produced from the grafted cellulose. The intermediate products and poly(hydroxamic acid) ligand were analyzed by FT-IR, FE-SEM, TEM, EDX, and XPS spectroscopy. The adsorption capacity (*q_e_*) of some toxic metals ions by the polymer ligand was found to be excellent, e.g., copper capacity (*q_e_*) was 346.7 mg·g^−1^ at pH 6. On the other hand, several metal ions such as cobalt chromium and nickel also demonstrated noteworthy sorption capacity at pH 6. The adsorption mechanism obeyed the pseudo second-order rate kinetic model due to the satisfactory correlated experimental sorption values (*q_e_*). Langmuir model isotherm study showed the significant correlation coefficient with all metal ions (R^2^ > 0.99), indicating that the single or monolayer adsorption was the dominant mode on the surface of the adsorbent. This polymer ligand showed good properties on reusability. The result shows that the adsorbent may be recycled for 6 cycles without any dropping of starting sorption capabilities. This polymeric ligand showed outstanding toxic metals removal magnitude, up to 90–99% of toxic metal ions can be removed from industrial wastewater.

## 1. Introduction

In the current era of globalization, the earth is confronted by abundant obstructive incident and challenges due to the anthropogenic activities for economic reasons. As a result, our planet is facing a diverse environmental problem, such as water pollution, intolerable solid waste generation, climate change, or global warming, resulting in significant environmental deterioration [1,2]. The effect of the environmental issues is horrifying on a global economic and social development scale. Heavy metals, one of the worst industrial pollutant, have been released at a large amount into the environment over the last few decades, and these heavy metals are not biodegradable [3]. Various types of industrial operation, e.g., electroplating, electrolysis depositions, milling, and metal coatings, etc., generated the harmful non-biodegradable toxic heavy metals in the factory sewage/wastewater [4]. Thus, adsorption investigators are given a wide attention and have made critical responses with different remediation techniques to address the issue of heavy-metals-contaminated residual discharges. Many countries have enacted strict regulations, environmental acts, and actions in view of this problem, resulting in a widespread practice of compelling industries to treat their heavy metals discharge before releasing it into the ecosystem [3]. The EPA in alliance with the WHO has set down the most reasonable level of toxic metals allowed in the environment to control the level and discharge of toxic metals in order to address the ecological, biological, and industrial concerns from industrial effluents. Adsorption, a phase transfer process, is a simple and effective process for wastewater treatment due to its simplistic nature of the adsorption process, which is easy to conduct with a low investment, and suitability to treat wastewater contains trace amount of heavy metals [5,6,7,8,9,10]. However, many adsorbents suffer from low adsorption capacity, selectivity issue, and difficulty to be regenerated making it an unsuitable choice for treatment process. Therefore, magnitude of interest has grown in the development of new adsorbent materials [11]. 

Absolutely, water is essential substance for all living elements, therefore, water has a vital role to the environment for the existence of life. However, water quality has been affected due to the urbanization, industrialization, agricultural activities, etc. Many types of water pollution and existence of toxic metals in the various types of water causes detrimental effects to the safe water [12,13,14,15]. An effective treatment method is needed for safe water to avoid the detrimental health effects caused by the toxic heavy metals. Physical treatment methods such as adsorption [16], precipitation [17], coagulation [18], reverse osmosis [19], and membrane filtration [20] have been conventionally used in industry processes. The adsorption by bio-adsorbents is also been validated as a promising and cost-effective method among other methods to get safe water [21]. Chemically modified cellulose can be a potential bio-adsorbent, consequently, chemical conversion of the cellulose might intensify its sorption performance as an efficient adsorbent [22].

The free radical polymerization method has been widely used to produce the grafting copolymers from various cellulose materials. Variety of the methods have been utilized to produce free radicals for polymerization reactions. However, ceric ions from ceric ammonium nitrate (CAN) can be a simple method to produce initiator radicals for grafting copolymerization. Certainly, the radicals are created on the cellulose (AGU, anhydroglucose unit) resulting in the formation of covalent bonds with the desired monomers; further, in many propagating units, the chains grow until termination process occurs. Ultimately, a grafting copolymer product can be generated via disproportionation step or combination of two growing chains of cellulose molecules [23,24,25]. Eventually, there is chemical conversion of the grafted cellulose into the chelating ligands specially hydroxamic acid for the coordination with metals ions in a simple binding system. In principle, atoms, molecules, or small groups of atoms exhibiting negatively or neutral charge, which may connect to a central metal atom, eventually form a coordination compound [26,27,28].

In this study, cellulose is extracted from waste fiber by alkaline solution for delignification. Then, the cellulose was used in grafting reaction with monomer to produce the poly(methyl acrylate)-grafted cellulose. Grafted cellulose was then transformed into the poly(hydroxamic acid) ligand for the sorption of toxic metals from synthetic water as well as electroplating wastewater. The spectroscopy analyses were used to perform the morphological and structural investigation of the intermediate and final products. Eventually, the adsorption capacity of poly(hydroxamic acid) ligand was evaluated using various parameter such as pH values, contact times, and isothermal study including the reusability study.

## 2. Materials and Methods

### 2.1. Extraction of Cellulose

Waste pandanus fruit fibers were obtained from Melinsung summer bay in Papar, Sabah, Malaysia. On the other hand, durian rinds were obtained from a random fruit market in Kota Kinabalu, Sabah. Waste pandanus fruit fiber and durian rinds were cut into small pieces and were ground. About 200 g of mixed waste fiber was boiled with 800 mL of sodium hydroxide (Merck, Germany) solution (18%) for 5 h. The fibers were washed with water and then boiled with approximately 550 mL of glacial acetic acid (Merck, Germany) for an hour followed by washing with water. After that, 400 mL of hydrogen peroxide (bleaching agent, Merck, Germany) was added into the cooking fibers followed by 400 mL of NaOH (8%), and it was boiled for 1 h. The resulting cellulose was washed 5 times with water and dried in oven at 50 °C. The dried cellulose was used in grafting process for modification of cellulose into grafting copolymer [29].

### 2.2. Graft Copolymerization

The copolymerization reaction was executed with a 1 L three-neck round-bottom flask immersed into thermostat water bath fixed with a stirrer and condenser. About 6.5 g of cellulose, 500 mL of distilled water, 1.1 mL of H_2_SO_4_ (50%), 1.1 g of ammonium ceric nitrate (10 mL H_2_O), and 10 mL of methyl acrylate (Merck, Germany) were heated at 55 °C with stirring under N_2_ atmosphere, and the reaction was carried out according to previous work [30]. The work-up procedure was also accomplished by previous work [30]. The poly(methyl acrylate)-grafted cellulose product was dried in an oven under 50 °C for 24 h.

### 2.3. Synthesis of Poly(hydroxamic acid) Ligand

Hydroxylamine (Merck, Germany) solution (300 mL) was made ready for oximation reaction according to the previous work [31]. About 10.0 g of poly(methyl acrylate)-grafted cellulose was added into a round-bottom flask with freshly prepared hydroxylamine solution at pH 10. The mixture was heated at 70 °C with stirring for 6 h. The work-up procedure was accomplished by previous work [23,25]. The resulting poly(hydroxamic acid) ligand was dried at 50 °C for 24 h.

### 2.4. Batch Adsorption Studies

A series of batch adsorption experiment was used to ascertain the metal ion binding capacity with various pH from 3 to 6 according to previous work [17,24]. Typically, 100 mg of poly(hydroxamic acid), 5 mL of distilled water, and 10 mL of 0.1 M of CH_3_COONa (Merck, Germany) buffer (pH 3 to 6) were placed into plastic bottles. Exactly 5 mL of 0.1 M of metal ions (copper sulfate-5H_2_O, cobalt chloride, chromium chloride, and nickel sulphate, all obtained from Merck, Germany)) was added into plastic bottles and stirred for 2 h. After the adsorption reached equilibrium, the metal cation concentration was analyzed by ICP-OES (Perkin Elmer OPTIMA 7300, Winter Street Waltham, MA 02451, USA). The metal ions concentration can be estimated by the following Equation (1):(1)$$q_{e}=\frac{\left(C_{o}-C_{e}\right)V}{L}$$
where *q_e_* stands for adsorption capacity (mg·g^−1^). *C_o_* and *C_e_* are the initial and equilibrium concentration of metal ions (mg·L^−1^), respectively. *V* is the volume of the metal ions solution (mL), and *L* is the mass of polymeric ligand (g).

The isothermal study was performed according to the previous work [31,32,33,34]. Similar batch adsorption process was carried out using the 100 mg of adsorbent, 10 mL of distilled water, and 5 mL of 0.1 M of CH_3_COONa buffer at pH 6, and exactly 5 mL of different concentrations of metal ions solution (10–2400 ppm) was shaken for 2 h. After equilibrium adsorption, the metal action concentration is analyzed by ICP-OES according to Equation (1).

The kinetic adsorption studies were also performed according to the previous work [35,36,37], batch adsorption was performed using the 100 mg of adsorbent, 5 mL of distilled water, 10 mL of 0.1 M of CH_3_COONa buffer solution at pH 6, and exactly 5 mL of 0.1 M of metal ion solution. The mixture was stirred using shaker machine at time intervals of 5, 15, 30, 60, and 120 min. Then, concentrations of metal cation were determined by ICP-OES, and calculation was performed using Equation (2):(2)$$q_{t}=\frac{\left(C_{o}-C_{t}\right)V}{L}$$
where *q_t_* is the quantity of metal ions adsorbed at time *t* (mg·g^−1^). Other parameters are expressed in Equation (1).

### 2.5. Desorption and Reusability Studies

For a typical desorption study, all metal ions were held at the adsorbent as a metal complexes. The metal ions were removed from the ligand using 2 M of HCl solution. The ligand was regenerated back by washing with water until remaining HCl was removed from the ligand. Then, CH_3_COONa buffer of pH 6 was added with every cycle of adsorption experiment for the reusability test for six cycles of adsorption-desorption process [25].

## 3. Results and Discussion

### 3.1. Synthesis of Poly(hydroxamic acid) Ligand

In this study, a monomer, namely, methyl acrylate was used in the grafting reaction onto cellulose at optimum conditions. According to the methods reported on the graft copolymer reaction, the mixture of cellulose, monomer, and an initiator in a round-bottom flask was constantly agitated and purged with nitrogen gas to create an inert condition in order to get rid of oxygen gas during graft copolymerization reaction [22,25]. The cellulose-grafted copolymer was then converted into chelating ligand, namely, poly(hydroxamic acid), and it was used as the adsorbents in the batch adsorption studies. The copolymerization reaction, depicted in Scheme 1, resulted to the poly(methyl acrylate)-grafted cellulose and further oximation reaction gave the poly(hydroxamic acid) ligand (Scheme 1). 

As expected, poly(hydroxamic acid) ligand was derived white in color (navajo white) as referred to previous studied [23,25]. The chelating ligand was prepared by the oximation reaction of acrylate functional groups containing esters of the poly(methyl acrylate)-grafted cellulose in basic condition. The reaction was performed in optimum conditions, whereby the mixture was constantly agitated under pH 10 for 4 h [25]. In this reaction, 10.00 g of cellulose-grafted copolymer yielded 12.11 g of poly(hydroxamic acid) ligand. During adsorption process, poly(hydroxamic acid) ligand bonded with Cu^2+^ ions to form poly(hydroxamic acid)-Cu complex by coordination system. Other metals-ligand complexes occurred by the same chelating ligand. The image of the poly(hydroxamic acid)-metal complexes formed by the coordination with copper, cobalt, chromium, and nickel is displayed in Figure 1, in which navajo white color of ligand turned into respective metal ions colors.

### 3.2. FT-IR Analysis

FT-IR spectra of waste cellulose, grafting copolymer, and poly(hydroxamic acid) are shown in Figure 2 (Perkin Elmer Spectrum 100, Winter Street, Waltham, MA, USA). FT-IR spectra obtained for a cellulosic material is shown in Figure 2a. A traceable peak at 886 cm^−1^ proved the C_1_-H deformation of glycosidic linkage [38]. A significant vibration for the of C-O-C pyranose ring showed stretching at 1152 cm^−1^. The stretching of C-H band appeared typically at 2907 cm^−1^, and an oblivious O-H stretching band was present at 3371 cm^−1^ as the cellulose structure contains six OH groups in each repeating unit. Besides, there were another two peaks shown at 1624 and 1408 cm^−1^, representing the bending modes of absorbed water and CH_2_ bonds, respectively.

IR spectra of the cellulose-graft-poly(methyl acrylate) are represented by the red-colored line in Figure 2b. A new prominent peak at 1715 cm^−1^, associated with the C=O stretching band was found as the grafted copolymer contains COOMe. There are two notable peaks rendered to the C-H bending scissoring by virtue of CH_2_ and wagging due to CH_2_ at 1452 and 822 cm^−1^, respectively [23,25]. In spite of the expected peak exhibited in the copolymer product, other peaks conserved for cellulose structure [25].

The blue-colored spectrum is representing the poly(hydroxamic acid) ligand (Figure 2c). A widen peak was detected at 3295 cm^−1^, which designated the existence of stretching band for N-H bonding belonging to the hydroxamic acid functional group. Consequently, as previously seen in Figure 2b, a widen peak for C=O stretching of the grafted copolymers at 1715 cm^−1^ was disappeared. Subsequently, loss of C=O stretching generated our anticipated bands at 1678 and 1653 cm^−1^, established for the hydroxamic acid functional groups, and both represented the C=O stretching and N-H bending, respectively. The spectra proved the successful conversion of the poly(hydroxamic acid) [23].

Eventually, the evidence of metal-ligand complexes is shown in Figure 2d for the copper-ligand coordination event occurred during adsorption process. The N-H band of the ligand (3295 cm^−1^) was changed by metal-ligand coordination system. Thus, the 1678 and 1653 cm^−1^ bands are associated with the stretching of C=N and N-H bonds respectively; both stretching bands are widen due to the formation of copper-ligand complexation, and this evidence is supported by the coordination of the copper ions by the polymer ligand [23].

### 3.3. FE-SEM Analysis

Characterization by the FE-SEM investigation was useful for some morphological analyses with JSM-7900F (JEOL, Tokyo, Japan). The SEM image of the pure cellulose extracted from waste fiber emerged as a smooth wooden stick-like structure as shown in Figure 3a. After the chemical modification of the cellulose into the grafting copolymer, the wooden stick-like structure was diminished, and unsmooth superficial morphology appeared for the grafted polymer in the SEM micrograph as shown in Figure 3b. No homopolymer lumps or distinguishable trace of material was found in the grafted copolymer. Further conversion of the grafted copolymer into the desired poly(hydroxamic acid) ligand changes morphological texture. As such, the unsmooth morphology of grafted copolymer disappeared, and unsmooth variable spherical beads were formed as shown in Figure 3c. Once polymer ligand complex was formed with the copper ions, then the spherical beads shrinker to form the smaller beads due to the coordination bonding that occurred between the ligand and copper ions (Figure 3d). The copper complex was also further analyzed by the TEM and EDX spectra.

### 3.4. TEM and EDX Analysis

Transmission electron microscope (TEM) examination was accomplished on Tecnai G2 Spirit BioTwin TEM (Dawson Creek Drive Hillsboro, USA) with a copper-grid-coated sample holder using 120 kV energy. TEM examination displayed the existence of copper nano complex onto waste cellulose-based poly(hydroxamic acid) surface, and average size of copper complex was estimated to be about ∅ = 21.7 ± 2 nm (Figure 4a). The copper nanocomplex showed small spherical morphology, and an irregular spreading of copper species was spotted with variable sizes. Further examination of the poly(hydroxamic acid)-Cu(II) complex was carried out through energy-dispersive X-ray (EDX) tool (JEOL Tokyo, Japan) (Figure 4b). Elements such as C, O, and N were mainly detected from the poly(hydroxamic acid, however, the presence of 35.4% of the copper species (8.03 keV) was determined, which was incorporated into the poly(hydroxamic acid)-Cu(II) complex.

### 3.5. X-ray Photoelectron Spectroscopy Analysis (XPS)

The X-ray photoelectron spectroscopy (PHI Quantera II, Physical Electronics, Lake Drive East, Chanhassen, MN, USA) was performed on the poly(hydroxamic acid) ligand and ligand-copper complex as shown Figure 5. The bidentate ligand, i.e., hydroxamic acid, strongly bonded with copper species, therefore, the XPS analyses can be most effective tools to determine the binding properties of copper with ligand. The survey scan XPS of poly(hydroxamic acid) displayed the binding energies (BEs) peaks at 284.1, 398.1, and 531.2 eV corresponding to 1s of carbon, nitrogen, and oxygen, respectively (Figure 5a). Poly(hydroxamic acid) ligand bonded with copper ions and formed a five-member Cu(II) complex that produced new BEs peaks at 935.4, 955.5, 125.3, and 80.4 eV corresponding to Cu2p3/2, Cu2p1/2, Cu3s, and Cu3p3, respectively (Figure 5b). In addition, the peaks belonging to the C 1s, N 1s, and O 1s were emerged with increased BEs at 286.2, 402.5, and 534.5 eV, respectively, due to decrease in electron density at ligand site (Figure 5b).

The narrow scans of oxygen and nitrogen atoms were performed for the confirmation of copper complex with hydroxamic ligand. The XPS spectra of poly(hydroxamic acid) ligand showed two peaks at the BEs of 530.5 and 532.2 eV, which corresponds to the O 1s in the O-NH and C=O groups, respectively (Figure 6a). However, after complexation of ligand with copper ions, the BEs of O 1s for O-NH increased from 530.5 to 531.0 eV and the O 1s for C=O was split into two peaks with increased BEs of 533.0 and 534.6 eV (Figure 6b). The peak appearing at BEs of 533.0 eV corresponds to O 1s for C=O and BEs of 534.6 eV belongs to the oxygen atom bonded with Cu (C=O-Cu) (Figure 6b).

On the other hand, we have performed another the core-level scan of 1s of nitrogen atom for the hydroxamic acid and its copper complex. In case of hydroxamic acid ligand, the BEs were at 398.9 and 400.6 eV corresponding to the N 1s for N-OH and N-H functional groups, respectively (Figure 7a). After complexation with copper, the BEs of N 1s for N-OH and N-H were increased to 399.6 and 402.8 eV, respectively (Figure 7b). After complexation, the BEs of N-OH and N-H were also shifted to higher BEs, which confirmed that the nitrogen atom participated in the complex formation with copper species. These higher BEs were observed as electron donation properties of oxygen and nitrogen atoms at the ligand site resulted in the decreased electron density [39].

### 3.6. Adsorption of Heavy Metal Ions

#### Effect of pH on the Adsorption

A distinguishable pH effect on the adsorption capacity of the metal ions by the polymeric ligands was found when compared with the previous work [23,25,40]. The pH effect on the adsorption behavior was studied using various metals (Cu^2+^, Co^2+^, Cr^3+^, and Ni^2+^) at room temperature with pH ranging from 3 to 6. A series of the batch adsorption was performed with sodium acetate buffers solutions (pH 3–6).

The adsorption capacity *q_e_* values were calculated as mg·g^−1^ based on Equation (1), and experimental data *q_e_* were plotted against various pH values (Figure 8). The *q_e_* was increased from pH 3 to 6, and optimum condition was found at pH 6 for all metal ions. Thus, copper (Cu^2+^) ions exhibited the highest binding ability as 346.7 mg·g^−1^ at pH 6, and other heavy metal ions showed adsorption capacities of 315, 227.6, and 181.4 mg·g^−1^ for Co^2+^, Cr^3+^, and Ni^2+^, respectively. A noticeable increase in the adsorption ability can be observed from pH 3 to 6, showing that the binding event of metal ions by poly(hydroxamic acid) was pH dependent in the order of Cu^2+^ > Co^2+^> Cr^3+^ > Ni^2+^. The presence of bidentate hydroxamate anions created five-membered ring complexes with metal ions [25].

The adsorption uptake action by the polymer adsorbent was achieved due to the presence of hydroxamic acid group -CH_2_(C=O)NH(OH) [30,32]. The binding event occurred due to low pH condition; the basic carbonyl group (-C=O) tends to be hindered from binding with the positively-charged heavy metal cations as protonation occurs at low pH as a result of higher concentration of H^+^, resulting in the formation of positively charged C=O^+^-H functional group. In addition, at low pH conditions, the ionization of acidic hydroxyl group (-OH) is unlikely to occur. These events result in reduced adsorption strength of heavy metal ions because at low pH conditions, there is relatively high concentration of acidic H^+^ cation, which competes with positively-charged heavy metal cations when adsorption occurs. Thus, adsorption uptake of heavy metal cations undergoes competition and they have lower affinity towards polymeric ligand at low pH conditions. On the contrary, in the acidic pH conditions (pH 5–6), the concentration of H^+^ ions is relatively in the lower range. As a result, the protonation of oxygen in the basic -C=O was less likely to occur. The coordination capability of carbonyl group of poly(hydroxamic acid) to metal ions was raised. Moreover, the ionization of acidic hydroxyl group (-OH) was increased and negatively-charged oxygen ions were formed, which favored the electrostatic interaction between the hydroxamic acid resin and the heavy metal cations, causing a higher adsorption strength of polymeric ligand towards heavy metal ions, and thus, ultimately resulted in the formation of metal-ligand complexes [27,28]. On the other hand, solutions with pH greater than 7 could not determine the significance of the study due to the chances of hydrolysis and precipitation of transition metal ions. At basic condition, the precipitation of transition metal ions in the hydroxide form is induced due to higher pH values (>7), therefore, concentration of metal ions can be lowered leading to the incorrect *q_e_* values.

### 3.7. Adsorption Kinetic Studies

In principle, adsorption process requires contact time by the adsorbent with the metal ions such as Cu^2+^, Co^2+^, Cr^3+^, and Ni^2+^. The kinetic (*q_t_*) adsorption capacity was determined for every interval period (5, 15, 30, 60, and 120 min), and ICP-OES was used for analysis of metal ions. Selected heavy metal ions were used for the kinetic study with poly(hydroxamic acid) ligand. Based on the kinetic study, the effect of pH on the sorption efficiency towards selected heavy metal ions was then determined at optimum pH condition (pH 6) for the complexation reaction [35]. Therefore, pH 6 was used for studying the adsorption kinetics along others parameter described in Figure 9.

Figure 9 displayed the interrelation between the different reaction contact time and the adsorption capacities of poly(hydroxamic acid) ligand towards various metal ions. From the adsorption kinetic study curves, it was found that all the metal ions showed similar trend with the adsorbent as a function of time (Figure 9). It showed that there is increase in adsorption capacities with increase in contact time from 5 to 120 min. Based on the graph, it can be seen that the uptake of Cu^2+^ ions by ligand exhibited the highest adsorption capacity at 120 min time interval (304.5 mg·g^−1^), while other heavy metal ions such as Co^2+^, Cr^3+^, and Ni^2+^ also showed considerably good adsorption capacities, which were 282.1, 209.1, and 178.0 mg·g^−1^, respectively.

As the time interval increased with adsorption kinetic studies via batch adsorption, agitation was required during the reaction as it increases the chances of uniform and even collision between the molecules of metal ions and the adsorbent. As a result, it increased the reaction rate. The adsorption mechanism involved the development of valance forces from the sharing and exchanging of electrons between the heavy metal ions involved and the poly(hydroxamic acid) ligand.

According to the graph obtained in Figure 9, the uptake of metal ions by adsorbent was at higher rate at the early time intervals. At initial time intervals of reaction, there were large numbers of the functional groups (-NHOH and -C=O) of poly(hydroxamic acid) ligand available to bind to the metal ions via coordination. The large number of active sites were available and free for binding of metal ions with the ligand [36]. The adsorption of metal ions by poly(hydroxamic acid) achieved equilibrium at 120 min. This indicated that the adsorption reaction rate after the time interval of 120 min was negligibly slow and eventually stagnant.

#### 3.7.1. Pseudo-First-Order Rate of Reaction

Determination of the rate of adsorption by the poly(hydroxamic acid) ligand involved either chemisorption or physisorption. Two common kinetic models, pseudo-first and pseudo-second-order rate of reactions, are utilized to study the adsorption kinetic behavior. Table 1 shows the pseudo-first-order rate of reaction model and its related kinetic parameters involved in the adsorption process.

Equation (3) was used to find out the kinetic behavior of pseudo-first-order rate of adsorption.
(3)$$\log(q_{e}-q_{t})=\log q_{e}-\left(\frac{K_{ads}}{2.303}\right)t$$
where *q_e_* and *q_t_* are the quantities of binding ions (mg·g^−1^) at equilibrium and *t* is the time (min) required for adsorption of metal ions. The pseudo-first-order rate constant (min^−1^) is expressed by *K_ads_*. The extent of parameters for *q_e_* and *K_ads_* were determined from the intercept and the slope of the graph (Figure 10) as log(*q_e_* − *q_t_*) versus time. The data obtained for the parameters with four metal ions are shown in Table 1. Based on the calculated data in Table 1, the R^2^ values of Cu, Co, Cr, and Ni were 0.8538, 0.8223, 0.8896, and 0.8643, respectively. The values are all insignificant (R^2^ < 0.89). On the other hand, the experimental adsorption (*q_e_*) and the calculated adsorption (*q_t_*) showed notable differences resulting in the pseudo-first-order rate of kinetic model being less fit to the experimental data.

#### 3.7.2. Pseudo-Second-Order Rate of Reaction

Equation (4) was used to find out the kinetic behavior of pseudo-second-order rate of adsorption.
(4)$$\frac{t}{q_{t}}=\frac{1}{k_{2}q_{e}^{2}}+\frac{t}{q_{e}}$$
where *q_t_* is the quantity of metal ions adsorption (mg·g^−1^) at a time interval and the *q_e_* is the metal ions adsorbed (mg·g^−1^) at equilibrium. In addition, *k*_2_ is the pseudo-second-order rate constant (g·mg^−1^ min).

A plot of *t*/*q_t_* against time and the corresponding values of *k*_2_ and *q_e_* were quantified from the graph (Figure 11). Based on the calculated results in Table 2, the R^2^ values of Cu, Co, Cr, and Ni were 0.9877, 0.9783, 0.9812, and 0.9920, respectively, and they are in acceptable ranges (R^2^ > 0.98). Of all the correlation coefficient values, R^2^ values were relatively large and showed high consistency. In addition, the experimental values (*q_e_*) of adsorption capacities of the metal ions were comparable with their respective calculated *q_t_* values. It can be observed that experimental adsorption (*q_e_*) and the calculated adsorption (*q_t_*) displayed minor differences of the pseudo-second-order rate of adsorption (Table 2). Therefore, kinetic data obtained from the graph suggested that the pseudo-second-order kinetic model was best fit with the experimental values. In conclusion, the pseudo-second-order rate of reaction kinetic was more predominant than pseudo-first-order kinetic. This kinetic behavior showed that the adsorption mechanism was chemically rate controlling [37,39]. Herein, the second-order mechanism is prevalent in the experimental data, therefore, the adsorption mechanism exhibited chemical process owing to the exchange of electrons between the metal ions and polymer ligand [39].

### 3.8. Adsorption Isothermal Studies

Commonly, the isothermal studies have been performed to find out the effect of initial concentrations of adsorbate on the adsorption capacity by the adsorbent. Therefore, a series of batch adsorption was performed using copper, cobalt, chromium, and nickel ions with various concentrations. In this study, except concentration of metal ions, other parameters were kept fixed as follows: 100 mg of poly(hydroxamic acid) ligands was used, pH of the solutions were set at pH 6, and agitation time was set to 2 h with constant speed. The concentration of the metal ions was kept variable such as 10, 300, 600, and 1200–2400 ppm. Figure 12 displayed the adsorption capacity of poly(hydroxamic acid) ligands against the four heavy metal ions at different heavy metal solution concentrations. As expected, that the adsorption capacity towards the heavy metals was increased with the increase in initial concentration of heavy metals, which eventually reached the equilibrium after 1200 ppm.

In the next subsection, we have studied some common isothermal studies for understanding the isothermal behaviors of the adsorption process. Therefore, most common isothermal models such as Langmuir isotherm model and Freundlich isotherm model were chosen to study the isothermal behaviors. Thus, common linear forms of Langmuir and Freundlich isotherm models were selected due to the mathematical simplicity of linear form model itself [32,33,34].

#### 3.8.1. Linear Langmuir Adsorption Isotherm

Langmuir adsorption isotherm model have been used to determine the monolayer adsorption; thus, Langmuir model is fit to the four criteria such as uniform surface and equivalent adsorption sites, no interaction of adsorbed molecules, adsorption mechanism is unique, and single layer or monolayer adsorption take place on the surface of the adsorbate [33,34]. The linear form of Langmuir adsorption isotherm is stated as Equation (5):(5)$$\frac{C_{e}}{q_{e}}=\frac{1}{q_{max}K_{L}}+\frac{C_{e}}{q_{max}}$$
where *C_e_* stands for the equilibrium metal ions concentration (mg·L^−1^), *q_e_* stands for the equilibrium adsorption capacity (mg·g^−1^) with corresponding metal ions solution, *K_L_* stands for Langmuir adsorption constant (L mg^−1^), and *q_max_* is the maximum capacity (mg·g^−1^) of the adsorbent.

Accordingly, the *q_e_* values were determined from the slopes of graph and Langmuir constant *K_L_* was calculated from the intercepts of the linear plots using the parameter *C_e_*/*q_e_* versus *C_e_* (Figure 13). Based on the data furnished in Table 3, all the correlation coefficient (R^2^) were significantly in accordance with Langmuir model (R^2^ > 0.99). The magnitudes of metals ions adsorption are in the order of Cu^2+^ > Co^2+^ > Cr^3+^ > Ni^2+^ corresponding to values of 357.1, 333.3, 238.1, and 185.2 mg/g, respectively (Table 3). This adsorption order was exactly similar with the order recorded in the previous pH and kinetic studies. In addition, the adsorption values calculated from Langmuir model showed very little differences with the experimental adsorption obtained previously. Since all of them were having R^2^ > 0.99, it was indicated that single or monolayer adsorption have been occurred on the surface of the adsorbent.

#### 3.8.2. Linear Freundlich Adsorption Isotherm

In principle Freundlich isotherm model have been used to rationalize the multiple layer adsorption in the heterogenous system of the adsorbent surface [34]. A linear Freundlich isotherm equation is conveyed in Equation (6).
(6)$$\log q_{e}=\log K_{F}+\frac{\log C_{e}}{n}$$

In Equation (6), the *q_e_* and *C_e_* stand for adsorption capacity (mg/g) and equilibrium adsorption capacity (mg·L^−1^), respectively, as described in previous equation. Here, *K_F_* is Freundlich’s constant, and 1/n is the heterogeneity factor.

The linear plots of Freundlich adsorption isotherm are used to find the parameters in the heterogenous system. Therefore, the heterogeneity factor n values were determined from the slopes of the graph, and the *K_F_* values were estimated from the intercepts of the graph using the plots of log *q_e_* versus log *C_e_* (Figure 14). The data associated with the graph are presented in Table 4. The overall data values of correlation coefficient were not well fitted (R^2^ < 0.96). In comparison with both isotherm models, Freundlich isotherm was less significant compared to that of Langmuir isotherm, indicating that it was unlikely to have multilayer adsorption on the heterogeneous surface of the poly(hydroxamic acid) ligand.

According to the calculated results presented in Table 5, it can be noticed that the Langmuir model had smaller HYBRID and MPSD values. Smaller HYBRID and MPSD values indicated smaller error in the estimation of *q_e_* values in the isotherm model. As a result, Langmuir model had more accurate and less error of estimating *q_e_* values, suggesting that Langmuir isotherm model was the best fit model and provided a better model of explaining the isothermal behavior of adsorption of heavy metal ions by the polymer ligand.

### 3.9. Reusability Study of Poly(hydroxamic acid) Ligand

Herein, the bio-waste materials were used as the source of cellulose materials to obtain the cellulose-based poly(hydroxamic acid) ligand; however, reusability study is essential to make sure its use in real-life applications. Reusability study comprehended the chemical process of elution and regeneration of the adsorbent. Based on the studies done on the pH effect on the adsorption by bio-adsorbent, it has been determined that the sorption of metal ions is unlikely to occur at very low pH as it is more prone to desorption process where the adsorbed metals can be desorbed out from the adsorbent [23,24]. Thus, by applying this concept, a 2 M of HCl solution was used to desorbed metal cations from the adsorbent [24]. Therefore, cobalt-poly(hydroxamic acid) complex was desorbed using HCl, and it was reused for a new adsorption cycle. The adsorption-desorption processes were performed for 6 cycles.

According to the adsorption capacity obtained earlier, the Co^2+^ optimum capacity was 315.0 mg·g^−1^, and as referred to this result, % sorption was calculated for each cycle. Based on the results displayed in Figure 15, the efficiencies of sorption and extraction based on the percentage of sorption and extraction data obtained were only having approximately 5% loss after 6 cycles of adsorption-desorption processes. According to the results, no significant loss of the adsorption capacity was found. In conclusion, the adsorbent indeed showed recyclable characteristic. Therefore, it is worthy to promote it in the removal of toxic metal ions from industrial wastewaters.

### 3.10. Practical Application of Poly(hydroxamic acid) Ligand

As determined by several parameters such as pH, contact time, concentration of adsorbate, etc., the study the practical application of the adsorbent for removal of metal ions in real-life application is simple. The ICP-OES results showed (Table 6) that the couple of metal ions were present in wastewater samples (IWS 1 and IWS 2) obtained from a semiconductor electroplating industry in Singapore. Table 6 and Figure 16 demonstrated that the quantity of metal ions in the wastewater samples (ppm) was diminished dramatically after treatment with the polymer adsorbent at optimum conditions. It was observed that the removal of metal ions was fruitful and noteworthy.

The metal ions elimination capability of Fe^3+^ was higher, i.e., its removal efficiency was more than 95%, while removal efficiency of Cu^2+^, Pb^2+^, and Zn^2+^ was approximately 80–90%. Other heavy metal ions such as Cr^3+^, Mn^2+^, and Ni^2+^ were also having removal efficiency of more than 60–70% for IWS 2. On the other hand, Cr^3+^ itself in IWS 2 had even more removal efficiency (>96%); in addition, Fe^3+^ and Cu^2+^ showed removal efficiency of 99% and 97 %, respectively, indicating that the elimination capability of heavy metal ions by poly(hydroxamic acid) ligand is potential. Several heavy metals including alkali and alkaline earth metals (Ba^2+^, Ca^2+^, Mg^2+^, K^+^, Rb^+^, Na^+^, Ag^+^, V^4+^, and Al^3+^) showed reduction of metal ions after treatment with the adsorbent. However, the removal was relatively lower and less significant compared to that of the removal of heavy metal ions. In summary of practical application, poly(hydroxamic acid) can be used as an effective adsorbent, which is the most practical way of removing toxic metal ions from the industrial wastewaters.

In principle, a ligand possessing a pair of non-bonding electrons belongs to the ions or molecules, which surrounds to the central metal atom to form a complex/coordination compound. Owing to the interest in toxic metal removal from wastewaters, a polymeric ligand, namely, the hydroxamic acid ligand, was derived from the modified waste fiber cellulose. Thus, a lone pair of electrons exists on the nitrogen and oxygen atoms of the hydroxamic acid groups that loses a lone pair of electrons, which co-ordinates with metal ions [23]. The cellulose moderation into the metal-binding ligands exhibits notable differences observed in the sorption magnitude. Various mechanisms of metal binding, including ion exchange, complexation, co-ordination/chelation, electrostatic interactions, etc., are known to us [25]. The binding properties of modified cellulose-based poly(hydroxamic acid) ligand is greatly affected by the types of metals and other conditions of sorption process [30,31].

## 4. Conclusions

The poly(hydroxamic acid) chelating ligand was synthesized by utilizing the extracted cellulose from waste fruit fibers, in which the durian fruit was edible but pandanus fruit was not edible. A series of adsorption studies such as effect of pH, reaction contact time (kinetic), isotherm study, ligand reusability, and hands-on usefulness of the ligand was performed. Based on this study, it is found that the adsorption of copper, cobalt, chromium, and nickel by the adsorbent is all pH dependent. According to the results of kinetic study obtained in this study, the adsorption process by the ligand followed pseudo-second-order rate of reaction compared to the pseudo-first-order rate of reaction. The isothermal studies performed in this study also showed that the adsorption isothermal behavior is best fit with Langmuir adsorption isotherm model (R^2^ > 0.99), suggesting that the synthesized poly(hydroxamic acid) ligand has homogenous surface and monolayer adsorption occurred on the ligand surface. Eventually, the practical metals waste showed excellent removal of metals, i.e., up to 90–99% of toxic metal ions was removed from industrial wastewater.

## Data Availability

The data presented in this study are available on request from the corresponding author.

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
