# Peer review of "Waste Fiber-Based Poly(hydroxamic acid) Ligand for Toxic Metals Removal from Industrial Wastewaterâ€"

_polymers, 2021, doi:10.3390/polym13091486_

Round 1
Reviewer 1 Report
Title, Abstract and conclusions are not clear. There is need to write about research gap in introduction.
There is need to enhance the review of literature by citing recent studies journal’s papers.
There is need to add the estimations, method and results.
There is need to add the discussion.
Language have to improvement
Author Response
Please see attached PDF file.

Reviewer 2 Report
Recommendation: Publish after major revisions noted.
Comments:
In the manuscript titled “Waste Fiber-based Poly(hydroxamic acid) Ligand for Toxic Metals Removal from Industrial Wastewater”, the authors chemically modified the pure cellulose extracted from the waste fiber to obtain, following graft polymerization reactions, the ligand poly (hydroxamic acid). The as-synthesized poly (hydroxamic acid) ligand was used for the adsorption of different heavy metal ions from wastewater. The article consists of a very preliminary set of results (FTIR, SEM, kinetic and isothermal adsorption of different heavy metal ions a various pH). I do recommend this paper to be published in Polymers, after a major revisions. The detailed comments are below:
- The authors reported in the manuscript that they obtained a yield of 121.1% in the final reaction of formation of the poly (hydroxamic acid) ligand. So please that, I ask the authors for more information on this unclear value.
- FTIR spectroscopy was used to study the chemical composition of the waste cellulose, grafted poly (methyl acrylate) cellulose, poly (hydroxamic acid) and poly (hydroxamic acid) copper complex, by analysing the presence followed by changes in the specific and key functional groups before and after adsorption of the copper ion. In particular, the copper-ligand complexation was confirmed by the widening of the 1678 and 1653 cm-1 bands associated with the stretching of C=N and N-H bonds. The authors should confirm this result with the study of the electronic structure, which is fundamental to investigate the electronic interactions (sorption mechanisms) between hydroxamic acid group -CH2(C=O)NH(OH) and the Cu ion, and, XPS represents the most suitable tool to carry out this task. In particular, the authors should investigate the possible variations of the XPS spectra in the binding energy regions of N 1s, O 1s and Cu 2p for the polymer ligand before and after copper adsorption, also showing the variation of atomic concentrations. In addition, I want to see the wide-scan XPS spectra of all systems investigated with FTIR.
- The authors studied the morphology of all systems using FE-SEM measurements. They stated that once polymer ligand complex with copper ions is formed, a change occurs in the morphology of the system, the spherical spheres have withdrawn to form the smaller pearls due to the coordination bonding takes place between the ligand copper ions. However, FE-SEM cannot be used to claim complexation has occurred unless supported by EDX measurements of the chemical composition. Therefore, the authors should show a TEM-EDX characterization of the poly (hydroxamic acid) ligand after Cu2+ adsorption to highlight the presence of the adsorbed copper ion on the system.
- The adsorption of different heavy metal ions was studied with pH ranged from pH 3 to 6. In the manuscript, the authors wrote that at low pH condition, the interactions between the hydroxamic acid group -CH2(C=O)NH(OH) and heavy metal cations are hindered due to the protonation of the carbonyl group (-C=O), resulting in formation of a positively charged functional group C=O+-H. As the pH (5-6) increases, the protonation competition of carbonyl oxygen -C=O is decreased, lower concentration of H+ ions in solution. Consequently, the coordination capacity of the carbonyl group of poly (hydroxamic acid) towards the metal ions has been increased thanks also to the synergy of the acid hydroxyl group (-OH). I would like to point out, in basic conditions this reaction should be more favoured. Why were the measurements not made at basic pH? Therefore, I would like to see at least this process at pH 7, which it is the real conditions under these heavy metal ions can be found in water.
Author Response
Please see attached PDF file.

Reviewer 3 Report
In current era of globalization, the earth is confronted by abundant obstructive incident and challenges, that generally caused by human and animals and others man-made activities. As a result, our planet faces number of environmental problems, such as water pollution, unsustainable solid waste generation, climate change or global warming, resulting in significant environmental deterioration over the long term. Undoubtedly, water is essential for all living elements, therefore, water has a vital role to the environment for the existence of life. However, water quality is affected due to the urbanization, industrialization and agricultural activities etc. Many types of water pollution, existence of toxic metals in the various types of water causes of detrimental effects to the safe water. Toxic metals in the industrial wastewaters have been liable a drastic pollution hence a powerful and economical treatment technology is needed for the water purification. For this reason, some pure cellulosic materials were derived from waste fiber. In this article, The poly(hydroxamic acid) chelating ligand was synthesized by utilizing the extracted cellulose from waste fruit fibre, in which this unique fruit is not common as public do not eat it although it is edible and safe to be consumed. A series of adsorption studied were performed such as effect of pH, reaction contact time (kinetic), isotherm, ligand reusability, and hands-on usefulness of the ligand were determined. The results and theme of this paper is quite interesting. The layout is clear and easy to understand. Generally, this manuscript makes fair impression and my recommendation is that it merits publication in this Journal, after the following major revision:
- The introduction should be reconstructed to present a coherent literature review. It may help the authors by answering the following questions: Why are these works relevant? Which specific problems were addressed? How are the previous results related with the latest work? What are the outstanding, unresolved, research issues? Answering the questions leads to the novelty of the proposed work naturally.
- In Fig. 4-10, the authors should give the explanations for the difference of data collected from different sources.
- Materials and Methods part. Although the results look “making sense”, the current form reads like a simple lab report. The authors should dig deeper in the results by presenting some in-depth discussion.
- On the other hand, several metal ions such as cobalt chromium and nickel also demonstrated noteworthy sorption capacity at pH 6. The adsorption mechanism obeyed the pseudo second-order rate of kinetic model due to the satisfactory correlated with the experimental sorption values (qe). Langmuir model isotherm study showed the significant correlation coefficient with all metal ions (R2 > 0.99). Since all of them showed R2 > 0.99 indicating the single or monolayer adsorption occurred on the surface of the adsorbent. This polymeric ligand showed outstanding toxic metals removal magnitude, up to 90-98% of toxic metal ions can be removed from industrial wastewater. The authors should give some explanation on above conclusions and data.
- Fibers have been widely used in the industry. The present work mainly focuses on lab work. It does not necessarily imply that the theoretic work (modeling) is not important. The authors omit this part during the current literature review, which should include a brief review of the theoretic work after revision. In the theoretic perspective, fractal theory is a very important tool, which can be used to investigate the adsorption mechanism, (see [A fractal model for capillary flow through a single tortuous capillary with roughened surfaces in fibrous porous media, Fractals, 2021, 29(1):2150017; Fractals, 2019, 27(7): 1950116]). Authors should introduce some related knowledge to readers. I think this is essential to keep the interest of the reader.
- Please, expand the conclusions in relation to the specific goals and the future work.
Author Response
Please see attached PDF file.

Round 2
Reviewer 1 Report
The findings are expressed clearly.
The conclusions and generalizations are based on the findings.
The language is clear and understandable.
The work is contributing to the field.
Reviewer 3 Report
It is ok.